# Emergence of IncHI2 Plasmid-Harboring blaNDM-5 from Porcine *Escherichia coli* Isolates in Guangdong, China

**DOI:** 10.3390/pathogens10080954

**Published:** 2021-07-29

**Authors:** Zhenbao Ma, Zhenling Zeng, Jiao Liu, Chang Liu, Yu Pan, Yanan Zhang, Yafei Li

**Affiliations:** 1Institute of Quality Standard and Monitoring Technology for Agro-Products, Guangdong Academy of Agricultural Sciences, Guangzhou 510640, China; mazhenbao@stu.scau.edu.cn; 2Guangdong Provincial Key Laboratory of Veterinary Pharmaceutics Development and Safety Evaluation, South China Agricultural University, Guangzhou 510642, China; zlzeng@scau.edu.cn (Z.Z.); 20182027012@stu.scau.edu.cn (J.L.); 20182027011@stu.scau.edu.cn (C.L.); pany@stu.scau.edu.cn (Y.P.); 3National Risk Assessment Laboratory for Antimicrobial Resistance of Animal Original Bacteria, South China Agricultural University, Guangzhou 510642, China; 4Institute of Animal Husbandry and Veterinary Medicine, Guizhou Academy of Agricultural Sciences, Guiyang 550005, China; 15285087157m@sina.cn

**Keywords:** *Escherichia coli*, *bla*_NDM-5_, IncHI2 plasmid, resistance

## Abstract

Carbapenem resistance has posed potential harmful risks to human and animals. The objectives of this study were to understand the prevalence of *bla*_NDM-5_ in pigs and investigate the molecular characteristics of NDM-5-producing *Escherichia coli* isolates in Guangdong province in China. Carbapenem-resistant *E. coli* isolates were isolated from pigs and obtained using MacConkey plates containing 0.5 mg/L meropenem. Conjugation assay and antimicrobial susceptibility testing were conducted for the isolates and their transconjugants. Whole-genome sequence (WGS) was used to analyze the plasmid genetic feature. A total of five *bla*_NDM-5_-carrying *E. coli* isolates were obtained in the present investigations. They belonged to five ST types. The *bla*_NDM-5_ genes were found to be in IncX3 and IncHI2 plasmid. The IncX3 plasmid was 46,161 bp in size and identical to other reports. IncHI2 plasmid was 246,593 bp in size and similar to other IncHI2-ST3 plasmids. It consisted of a typical IncHI2 plasmid backbone region and a multiresistance region (MRR). The *bla*_NDM-5_ was closely associated with the *IS3000-ISAba125-blaNDM-5-bleMBL-trpF-tat-IS26* unit. We first reported the *bla*_NDM-5_-carrying IncHI2 in *E. coli* isolates recovered from pigs and revealed the molecular characterization. Continued surveillance for the dissemination of *bla*_NDM-5_ among food-producing animals is required.

## 1. Introduction

Carbapenems are considered to be the most active and potent agents in the treatment of severe infections caused by multidrug-resistant (MDR) Gram-negative pathogens [1,2]. In the past few years, the emergence and spread of carbapenem-resistant *Enterobacteriaceae* (CRE) has led to a variety of infectious diseases with high mortality rates, which has posed a critical threat to clinical treatment and a public health concern [3,4]. New Delhi metallo-β-lactamase (NDM) is a type of metallo-β-lactamase (MBL) that hydrolyzes most carbapenems except for monobactams [5]. Since the *bla*_NDM-5_ gene was first reported, it has been the most prevalent carbapenem resistance gene in *Enterobacteriaceae* from humans and animals as demonstrated by several epidemiological investigations published in recent years [6,7]. 

Plasmids considered to be an important vehicle, play a vital role in the dissemination of the resistance genes. The *bla*_NDM-5_ gene is reported to be located on a variety of plasmid replicon types, such as IncFII-IncFIB [8], with IncX3 being the major carrier [9,10,11]. Recently, *bla*_NDM-5_ gene was found from the surrounding via IncX3 plasmid. These findings warn the wide-dissemination possibility of carbapenem resistance gene and emphasize the importance of monitoring the prevalence. Herein, the aim of our study was to investigate the prevalence of *bla*_NDM-5_-positive *Escherichia coli* strains from pigs in Guangdong province, China, and revealed the *bla*_NDM-5_-carrying plasmid in *E. coli* isolates recovered from pigs. 

## 2. Results

In this study, five isolates positive for *bla*_NDM-5_ gene from one swine farm were obtained. The susceptibility test indicated that five isolates were phenotypically resistant to β-lactams and displayed patterns of MDR, such as ampicillin, ceftazidime, meropenem, apramycin, florfenicol, sulfamethoxazole/trimethoprim, but sensitive to gentamycin, amikacin, colistin, and tigecycline, as shown in Table 1. According to PFGE analysis, five isolates exhibited five different PFGE patterns, designated A to E (Figure 1). Considering the source of isolates, we presumed that the dissemination of the *bla*_NDM-5_ gene may be due to horizontally transfer. 

Based on the whole-genome sequence analysis, the MLST of five isolates were divided into five distinct sequence types, namely ST10, ST48, ST155, ST2937, and ST4063. The isolates harbored numerous antimicrobial resistance genes such as *bla*_TEM-1B_, *bla*_OXA-10_, *bla*_CTX-M-14_, *aadA2*, *cmlA1* and *floR*, as seen in Figure 1. These resistance genes were associated with antibiotic-resistant phenotype. The five *bla*_NDM-5_-positive isolates were successfully transferred the *bla*_NDM-5_-5 gene to *E. coli* J53 at frequencies of 0.15 × 10^−6^ to 5.98 × 10^−6^ transconjugant/recipient (Table 1). The transconjugants were resistant to ampicillin, cefoxitin, ceftazidime, meropenem, florfenicol, and sulfamethoxazole/trimethoprim (Table 1).

It is noteworthy that *bla*_NDM-5_ genes were in two replicon types of plasmids. After we analyzed the genomic features of these five strains by the next-generation sequencing, we found two types of plasmids in the *bla*_NDM-5_-positive strains. PCR-based replicon typing indicated that transconjugants GDB8P64J and GDB8P70J carried the IncHI2 plasmid replicon and exhibited the same MDR phenotype, while the remaining transconjugants carried the IncX3 plasmid replicon and only displayed resistance to β-lactams. A complete IncX3 plasmid carrying the *bla*_NDM-5_ gene was detected in three isolates (GDB8P65M, GDB8P75M and GDB8P77M) according to PCR mapping and the primers listed in Table 2. Sequence analysis indicated that the IncX3 plasmid pHNGD75-NDM (accession numbers MT296100) was 46,161 bp in size and was almost identical to the IncX3 plasmids found in *Enterobacteriaceae* such as pCNUH-14 (*E. coli*, MK986791) and pNDM5_IncX3 (*K. pneumoniae*, KU761328). 

Sequence analysis also demonstrated that the *bla*_NDM-5_ gene was in the IncHI2 plasmid, named pHNGD64-NDM (accession numbers MW296099). Plasmid pHNGD64-NDM belonged to IncHI2-ST3. It was 246,593 bp in size and similar to other IncHI2-ST3 plasmids, it consisted of a typical IncHI2 plasmid backbone region and a multiresistance region (MRR). The backbone region of plasmid pHNGD64-NDM contained a typical IncHI2 plasmid backbone region associating with plasmid replication, conjugal transfer, and maintenance and stability, as well as a set of tellurite resistance determinants. Its similarity to those *bla*_NDM-5_-carrying IncHI2-ST3 plasmids such as plasmids p8C59-NDM (*E. coli*, chicken, MT407547), pNDM33-1 (*E. coli*, duck, MN915011) and pNDM-TJ33 (*E. coli*, duck, MN915010) (Figure 2). Furthermore, the MRR of pHNGD64-NDM contained several antibiotic resistance genes (e.g., *addA1*, *cmlA5*, *aph(4)-Ia*, *bla*_OXA-10_, *bla*_NDM-5_, *arr-2*, *drfA14* and *floR*) and insertion sequences (e.g., IS*26*, *Tn2*, IS*Ec59*, IS*3000*, IS*5*, *Tn5359*, IS*Kpn19*, IS*1006*, IS*5075*, and *Tn21*). The MRR of pHNGD64-NDM was inserted into the downstream region of hipB-A in accordance with plasmids p8C59-NDM and pNDM33-1. However, it was distinct from IncHI2 plasmid p8C59-NDM and displayed high similarity to that of pNDM33-1, which differed in its loss of the segments containing the ~11.8-kb genetic fragment (*IS3000-hp-IS26-lnu(F)-**∆intI-IS26-IS26-**∆Tn3*) and a 5489-bp *tetA* module (*∆TnAs1-tetA-tetR*) (Figure 1). Moreover, a fragment with a size of 32,661-bp including a class I integron cassette and blaNDM-5 module in pHNGD64-NDM showed >99% homology with plasmids pNDM33-1 and pNDM-TD33, in addition, a fragment was observed with the size of 41,837-bp containing a *floR* module (*∆ISCR2-floR-**∆ISCR2*), *qnrS* module (*IS26-qnrS-ISKpn19*) and an incomplete class I integron harboring the *arr-2| cmlA5| blaOXA-10| aadA1| drfA14|* cassette array, which was truncated by two copies of *IS26* with the same direction, but the direction of the *qnrS* module in pHNGD64-NDM was the opposite (Figure 3).

## 3. Discussion

Plasmid-mediated horizontal transmission of drug-resistant genes has been recognized as an important pathway for the rapid transmission of drug-resistant genes in *Enterobacteriaceae*, of which IncX3, IncFII and IncHI2 plasmids play a key role in the transmission of carbapenem-resistant *bla*_NDM-5_ genes in Gram-negative bacteria [12]. Until now, *bla*_NDM-5_ gene has been reported to be prevalent in various members of *Enterobacteriaceae*. Recently, plasmid-encoded *bla*_NDM-5_ gene was found in *Salmonella typhimurium* of pork origin [13,14] and *Klebsiella pneumonia* [4,15] via IncX3 plasmids. IncX3 is a group of narrow-host-range plasmid, with the size of 46,161 bp, and an important transmission vector for *bla*_NDM-5_ gene, with a variety of *bla*_NDM-5_ subtypes, such as *bla*_NDM-1_, *bla*_NDM-4_, *bla*_NDM-5_, *bla* _NDM-7_ and *bla*_NDM-21_ [16,17]. IncX3 plasmid has been reported in many regions of the world, and has been widely reported in clinical human medicine, food animals and the environment [9]. In this study, three strains of *bla*_NDM-5_-positive *E. coli* were detected in IncX3 plasmid. Genomic sequence analysis showed that the IncX3 plasmid was highly similar to that reported in the *Enterobacteriaceae* such as *E. coli* and *Klebsiella pneumoniae* from human clinic, environment and food animals published in the database [18,19], suggesting that IncX3 plasmid has been widely prevalent.

In this study, for the first time, we detected and revealed the *bla*_NDM-5_-carrying IncHI2 in *E. coli* isolates recovered from pigs. Although the sample number is small and only two strains carry the IncHI2 plasmid, the presence of this plasmid should not be ignored. IncHI2 is a group of low-copy number, wide-host plasmid, which was widely distributed in *Enterobacteriaceae*. IncHI2 plasmid were described to carry a variety of drug-resistant genes, such as *oqxAB*, *bla*_CTX-M_, *bla*_NDM_, *bla*_KPC_, *fosA3* and *mcr-1* gene [13,20,21,22]. Reports on the *bla*_NDM-5_-carrying IncHI2 plasmid are scarce. Previous reports have also suggested that the IncHI2 plasmid was considered to be an important vehicle in the spread of carbapenem resistance genes such as *bla*_NDM-1_, *bla*_NDM-4_, *bla*_NDM-9_ and *bla*_IMP-4_ among *Enterobacteriaceae* from human and animals [13,23,24]. Since the discovery of *mcr-1* gene in 2015 that threatens the effectiveness of polymyxins, the IncHI2 plasmid that carried *mcr-1* has been found in *Enterobacteria* from a variety of sources, including humans [25], animals (cattle, pigs, chickens, ducks and turkeys) [26,27], food (chicken, beef, pork, milk and vegetables) [28,29] and the environment (farm soil and wastewater) [30] in several countries around the world. In this study, IncHI2 plasmid was co-transmitted with many drug resistance genes such as *floR, cmlA, sul1, acc(3)-IV, aph(4) -Ia* and *arr-2*, which undoubtedly exacerbated the resistance crisis of animal borne bacteria. In addition, after carefully looking up relevant information in the NCBI database, IncHI2 plasmids carrying *bla*_NDM-5_ gene were found in *E. coli* from chicken, ducks, and pigs in Guangdong province, but there were few reports in other regions. The present findings suggest that IncHI2 plasmid has become an important transmission vector for *bla*_NDM-5_ gene in food animals in Guangdong province. This type of plasmid may have spread in different areas of Guangdong province, and adaptive evolution occurred in the variable region of the plasmid during transmission. In this study, the sequence of porcine *E. coli* plasmid pNGD64-NDM showed high homotypes with other IncHI2 plasmid such as pNDM33-1 obtained in the duck farm that uploaded by our teaching and research department. We speculate that this phenomenon may be related to the large number of copies of insertion sequence *IS26* in the variable region, which lead to the variation of the variable region and then the inversion, deletion, and insertion of some sequences. It is noteworthy that the 9525-bp *bla*_NDM-5_ module was found in various replicon plasmids such as IncHI2 plasmids p8C59-NDM and pNDM33-1, IncI-γ plasmid pNDM-TD33 and IncX3 plasmid pHNGD75-NDM, which suggests that the dissemination of *bla*_NDM-5_ was closely associated with the IS*3000*-IS*Aba125-bla*_NDM-5_-*ble_MBL_-trpF-tat-*IS*26* unit.

## 4. Materials and Methods

### 4.1. Bacterial Collection, Species Identification, and Molecular Detection of Carbapenemase Genes

A total of 357 fecal samples were collected from three swine farms in two cities (Maoming and Qingyuan) in Guangdong province from May to November in 2018, with the sample of 180, 81, and 96 in each farm, respectively. Carbapenem-resistant *E. coli* isolates were screened using MacConkey plates containing 0.5 mg/L meropenem and only one red colony was selected from each fecal sample. Meropenem-resistant isolates from these swine farms were identified as *E. coli* by MALDI-TOF-MS. PCR and sanger sequencing were further done to confirm whether the resistant strains harbored the *bla*_NDM-5_ gene. The primer for the *bla*_NDM-5_ was conducted to confirm the presence of *bla*_NDM-5_ gene as previous report [31].

### 4.2. Conjugation and Antimicrobial Susceptibility Testing

To determine the transferability of *bla*_NDM-5_-bearing isolates, the transfer frequency for each strain was determined using the broth mating method, with sodium-azide-resistant *E. coli* J53 used as the recipient strain. The donor strains and the recipient strain were mixed at the ratio of 1:4 in Luria-Bertani (LB) broth, and then incubated at 37 °C for 4 h, the transconjugants were selected on MacConkey agar plates supplemented with 0.5 mg/L meropenem and 200 mg/L sodium azide and then confirmed by PCR method.

The minimum inhibitory concentrations (MICs) of 15 antimicrobial drugs against the *bla*_NDM-5_ positive isolates and their transconjugants were determined using the agar dilution method or broth microdilution method (colistin) and values were then interpreted according to the document of CLSI-2020. Breakpoints of neomycin (>8 mg/L) and florfenicol (>16 mg/L) were interpreted according to EUCAST (http://mic.eucast.org/Eucast2/, accessed on 24 September 2020). The following antimicrobial agents were used in the present research: ampicillin, cefoxitin, ceftazidime, meropenem, gentamycin, amikacin, neomycin, apramycin, doxycycline, tigecycline, florfenicol, colistin, ciprofloxacin, enrofloxacin and sulfamethoxazole/trimethoprim. *E. coli* ATCC25922 served as the quality control strain.

### 4.3. Molecular Analysis of bla_NDM-5_-Positive Isolates

The genetic typing of *bla*_NDM-5_-positive *E. coli* isolates was digested with restriction endonuclease Xbal and conducted by pulsed-field gel electrophoresis (PFGE) according to our previous study [32]. The band patterns were analyzed with BioNumerics software version 5.10 (Applied Math s, Austin, TX, USA). Cluster analysis of fingerprinting similarity was conducted on software MasterScripts v4.0. The bacterial DNA of *bla*_NDM-5_-positive *E. coli* isolates were extracted by the DNA Extraction Kit, and then were sent to laboratory by next-generation sequencing. The multi-locus sequence type (MLST) of *E. coli* was performed by *Escherichia* typing database (https://pubmlst.org/bigsdb?db=pubmlst_escherichia_seqdef, accessed on 10 September 2020).

### 4.4. Whole-Genome Sequence and Plasmid Analysis

The *bla*_NDM-5_-carrying plasmid pHNGD64-NDM was sequenced using PacBio platform and assembled using HGAP version 4.0. to analyze the genetic feature. BRIG software was used to comparative analysis with other plasmid sequences published in NCBI (Genbank accession number MT407547, MN915011, MN915010). The complete genome of plasmid was annotated and analysis using RAST (https://rast.nmpdr.org/rast.cgi, accessed on 21 September 2020), ISfinder (https://www-is.biotoul.fr/, accessed on 12 September 2020), Resfinder (https://cge.cbs.dtu.dk//services/ResFinder/, accessed on 12 September 2020) and Vector NTI program (Invitrogen).

## 5. Conclusions

To the best of our knowledge, this study was the first to identify the *bla*_NDM-5_-carrying IncHI2 recovered from porcine *E. coli* isolates. Although carbapenems have not been approved for use in food-producing animals, multiple resistance genes were observed in the IncHI2 plasmid pHNGD64-NDM in the present research, which suggests that the spread and evolution of the *bla*_NDM-5_ gene may be selected and accelerated under the selective pressure of other antibacterial drugs. Thus, the dissemination of *bla*_NDM-5_ among food-producing animals requires continued surveillance.

## Figures and Tables

**Figure 1 pathogens-10-00954-f001:**
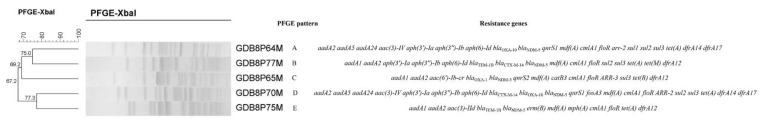
PFGE, antibiotic resistance genes of five porcine *bla*_NDM-5_-producing *E. coli* isolates.

**Figure 2 pathogens-10-00954-f002:**
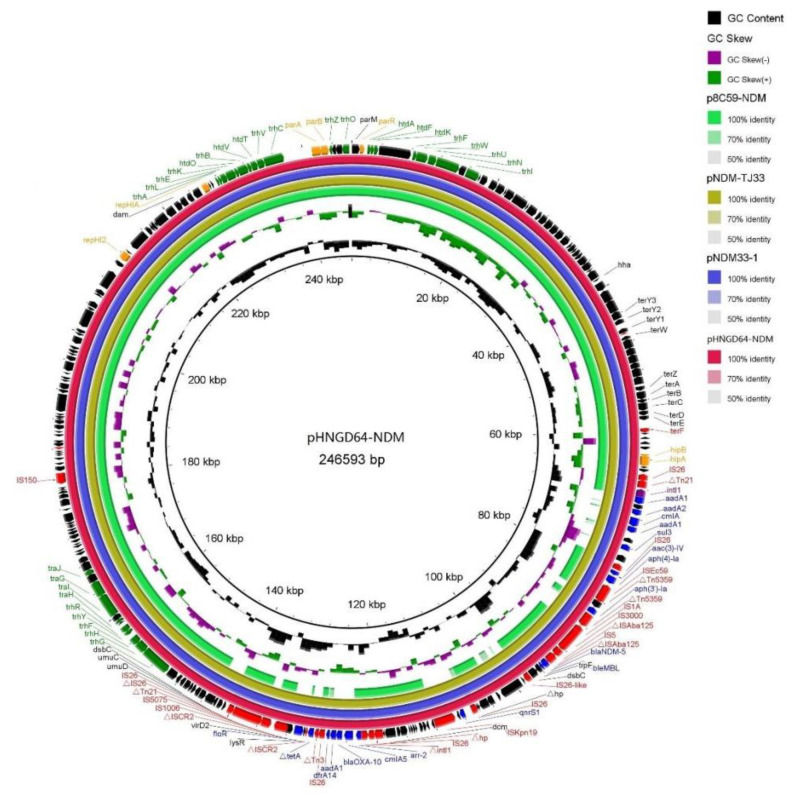
Ring comparison of IncHI2 plasmid pHNGD64-NDM using BRIG. Arrows indicate the positions and directions of gene transcription. ∆Indicates a truncated gene. Blue, red, green and black arrows indicate resistance genes, insertion sequences and transposes, transfer-associated protein, and hypothetical protein.

**Figure 3 pathogens-10-00954-f003:**
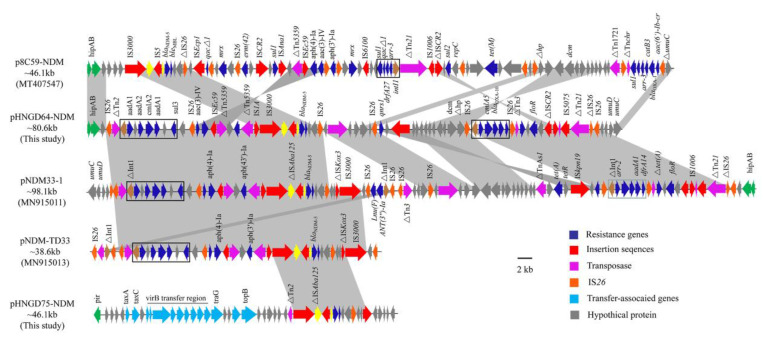
Liner comparison of the MRR region of plasmid pHNGD64-NDM. Arrows indicate the positions of the genes and the direction. Regions with >99% homology are shaded in gray. ∆indicates a truncated gene or mobile element.

**Table 1 pathogens-10-00954-t001:** Characterization of *bla*_NDM-5_-producing *E. coli* and their transconjugants.

Strain	GDB8P64	GDB8P64J	GDB8P65	GDB8P65J	GDB8P70	GDB8P70J	GDB8P75	GDB8P75J	GDB8P77	GDB8P77J	*E. coli* J53	*E. coli* 25922
MLST ^a^	ST4063	-	ST10	-	ST2937	-	ST48	-	ST155	-	-	-
Plasmid type ^b^	IncHI2 IncFIB IncQ1 p0111	IncHI2	IncX3 IncY	IncX3	IncHI2 IncFIB IncFII IncY	IncHI2	IncFIA IncR IncFII IncX3	IncX3	IncFIB IncN IncX3 p0111	IncX3	-	-
Transfer frequencies ^c^	5.70 × 10^−6^	-	0.15 × 10^−6^	-	5.98 × 10^−6^	-	2.32 × 10^−6^	-	0.68 × 10^−6^	-	-	-
MIC ^d^												
AMP	>128	>128	>128	>128	>128	>128	>128	>128	>128	>128	4	4
CTX	>64	>64	>64	>64	>64	>64	>64	>64	>64	>64	0.06	0.03
CAZ	>64	>64	>64	>64	>64	>64	>64	>64	>64	>64	0.06	0.06
MEM	8	2	16	4	8	2	16	4	16	4	0.016	0.016
GEN	4	0.5	0.5	0.5	4	0.5	0.5	0.25	0.5	0.25	0.25	0.25
AMI	1	0.5	4	0.25	2	0.5	2	0.5	2	0.5	0.5	0.5
NEO	64	0.5	1	0.5	64	0.5	1	0.5	32	1	0.5	1
APR	>128	1	4	1	>128	1	4	1	4	1	1	1
DOX	32	0.25	64	0.5	32	0.5	64	0.5	32	0.5	0.5	1
TIG	1	0.06	1	0.06	1	0.06	1	0.06	0.5	0.06	0.06	0.06
FLR	128	64	>128	2	>128	128	>128	2	>128	4	2	2
CL	0.25	0.125	0.25	0.125	0.25	0.125	0.25	0.125	0.25	0.125	0.125	0.125
ENR	32	0.5	1	0.016	16	0.5	2	0.016	16	0.016	0.016	0.008
CIP	8	0.25	0.5	0.008	8	0.25	2	0.008	8	0.004	0.008	0.008
SXT	>64/1216	4/76	>64/1216	<0.25/4.75	>64/1216	8/152	>64/1216	<0.25	>64/1216	<0.25/4.75	<0.25/4.75	<0.25/4.75

Note: ^a^: letter “J” represents transconjugants. ^b^: The multi-locus sequence type (MLST) of *E. coli* was performed by Escherichia typing database (https://pubmlst.org/bigsdb?db=pubmlst_escherichia_seqdef, accessed on 10 September 2020). ^c^: Plasmid type of Enterobacteriaceae was determined using CGE online website (https://cge.cbs.dtu.dk/services/PlasmidFinder/, accessed on 12 September 2020). ^d^: Transfer frequencies were calculated as the ratio of transconjugants over recipient cells, value was average mumble at least three times. Abbr. AMP: ampicillin, CTX: cefoxitin, CAZ: ceftazidime, MEM: meropenem, GEN: gentamycin, AMI: amikacin, NEO: neomycin; APR: apramycin, DOX: doxycycline, TIG: tigecycline, FLR: florfenicol, CL: colistin, CIP: ciprofloxacin, ENR: enrofloxacin and STX: sulfamethoxazole/trimethoprim.

**Table 2 pathogens-10-00954-t002:** Primer sequences of PCR mapping for *bla*_NDM_ gene.

Name	Prime Sequence (5′ to 3′)	Target Fragment	Reference
umuC-tat	F:GCGTAGCGTTTCCATAGCGG	1912 bp	This study
	R:GTTGACGGGTCTTTGGTGCT		
∆Tn2-hp	F:TGAAATGGCATGGGAATGAG	1294 bp	This study
	R:TTTCTGCGACAGTGATAGCG		
	R:GCTTTTGAAACTGTCGCACCT		

Note: F: forward primer; R: reverse primer.

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
