# Peer review of "Emergence of IncHI2 Plasmid-Harboring blaNDM-5 from Porcine Escherichia coli Isolates in Guangdong, China"

_pathogens, 2021, doi:10.3390/pathogens10080954_

Round 1

Reviewer 1 Report

In this study Ma et al. present evidence regarding carriage by pigs raised in the Guangdong province, China of diverse E. coli strains carrying IncX and IncHI2 plasmids encoding an MBL type, NDM-5. This finding is of public health interest given the importance of this carbapenemase. Indeed, the data presented here along with already publish findings relevant to the epidemiology of NDM-5-producers may indicate a significant contribution of swine farming in the spread of these pathogens. Also, experimental methodology and analyses of sequences are sound.

I must point out, however, that some points need clarification.

Here are my comments:

  1. There is no information about the units (e.g. average number of animals). Please include the relevant information
  2. The number of samples collected during six months (n=357) is likely too low to obtain quantitatively valid conclusions. This limitation should be briefly discussed.
  3. Lines 16-17 and lines 171-172. In the Abstract it is indicated that the carbapenem resistant E. coli were isolated using MacConkey agar supplemented with 0.5mg/L meropenem. However, in the Materials and Methods section the screening on MacConkey agar supplemented with 1mg/L meropenem is mentioned. Please check and correct.
  4. Lines 133-136. Please add a reference to support this statement.

Reviewer 2 Report

The manuscript presents the results of a study conducted on five strains isolated from faecal samples, including their characterisation and interestingly, a conjugation experiment to demonstrate the phenotypical acquisition of resistance features in addition to the whole-genome sequencing report. The experimental design is well presented, along with the results. It is not clear if the study included the WGS of the five strains, in this case, this must be clarified and methods need to be included along with more details on the genomic features and the results could be presented in a table as supplemental material.

Minor suggestions:

Line 59: remove "next"

L181-182: include the description of the method, reagents and devices.

L195: include reference of the mentioned study

Author Response

This manuscript is a resubmission of an earlier submission. The following is a list of the peer review reports and author responses from that submission.